# Paradoxical resistance of multiple myeloma to proteasome inhibitors by decreased levels of 19S proteasomal subunits

Diego Acosta-Alvear[1,2†], Min Y Cho[1,2,3], Thomas Wild[4], Tonia J Buchholz[5], Alana G Lerner[5], Olga Simakova[6], Jamie Hahn[6], Neha Korde[7,8], Ola Landgren[7,8], Irina Maric[6], Chunaram Choudhary[4], Peter Walter[1,2*], Jonathan S Weissman[2,3*], Martin Kampmann[3,2†‡]

[1]Department of Biochemistry and Biophysics, University of California, San Francisco, San Francisco, United States; [2]Howard Hughes Medical Institute, University of California, San Francisco, San Francisco, United States; [3]Department of Cellular and Molecular Pharmacology, University of California, San Francisco, San Francisco, United States; [4]The Novo Nordisk Foundation Center for Protein Research, University of Copenhagen, Copenhagen, Denmark; [5]Onyx Pharmaceuticals, Inc. an Amgen subsidiary, South San Francisco, United States; [6]Department of Laboratory Medicine, Clinical Center, National Institutes of Health, Bethesda, United States; [7]Multiple Myeloma Section, Lymphoid Malignancies Branch, National Cancer Institute, Bethesda, United States; [8]Myeloma Service, Department of Medicine, Memorial Sloan Kettering Cancer Center, New York, New York, United States

*For correspondence: peter@ walterlab.ucsf.edu (PW); Jonathan.Weissman@ucsf. edu (JSW)

†These authors contributed equally to this work

Present address: ‡Department of Biochemistry and Biophysics and Institute for Neurodegenerative Diseases, University of California, San Francisco, San Francisco, United States

**Abstract** Hallmarks of cancer, including rapid growth and aneuploidy, can result in non-oncogene addiction to the proteostasis network that can be exploited clinically. The defining example is the exquisite sensitivity of multiple myeloma (MM) to 20S proteasome inhibitors, such as carfilzomib. However, MM patients invariably acquire resistance to these drugs. Using a next-generation shRNA platform, we found that proteostasis factors, including chaperones and stress-response regulators, controlled the response to carfilzomib. Paradoxically, 19S proteasome regulator knockdown induced resistance to carfilzomib in MM and non-MM cells. 19S subunit knockdown did not affect the activity of the 20S subunits targeted by carfilzomib nor their inhibition by the drug, suggesting an alternative mechanism, such as the selective accumulation of protective factors. In MM patients, lower 19S levels predicted a diminished response to carfilzomib-based therapies. Together, our findings suggest that an understanding of network rewiring can inform development of new combination therapies to overcome drug resistance.

## Introduction

Protein degradation by the ubiquitin-proteasome system (UPS) fulfills essential roles in eukaryotic cells in maintaining proteome homeostasis (proteostasis), signaling, and cell cycle progression. The 26S proteasome is a large macromolecular machine composed of the 20S catalytic core and the 19S regulator, each comprised of more than a dozen different protein subunits. The 19S proteasome regulator binds polyubiquitinated proteins and catalyzes their deubiquitination and delivery to the 20S proteasome core for proteolysis. The 20S core proteasome can also associate with alternative proteasome regulators, such as the 11S complex (*Nathan et al., 2013*; *Schmidt and Finley, 2014*).

**eLife digest** Cells have several mechanisms for removing proteins that have been damaged or are no longer needed. One of these mechanisms is carried out by a large protein complex called the proteasome. Drugs that block the proteasome are toxic to all cells, and a type of blood cancer called multiple myeloma is particularly sensitive to these 'proteasome inhibitors'. However, tumors in patients with multiple myeloma can also become resistant to these drugs.

Using a genetic approach, Acosta-Alvear et al. identified the factors that control the sensitivity of cells to proteasome inhibitors. In particular, reducing the levels of other factors that contribute to protein balance made the cells more sensitive. Using a combination of proteasome inhibitors and drugs that target these other factors could prove to be useful in the fight against multiple myeloma.

The proteasome complex contains two types of subunits: regulatory subunits that recognize the proteins that need to be degraded, and catalytic subunits that degrade the proteins. The results of Acosta-Alvear et al. revealed how varying the levels of these two subunits influenced the sensitivity of cells to inhibitors. While decreasing the levels of catalytic subunits made the cells more sensitive, as expected, decreasing the level of regulatory subunits surprisingly made the cells resistant to the inhibitors. A possible explanation for this paradoxical result is that certain proteins are less effectively degraded by the proteasome in these cells, and that the buildup of these proteins protects the cells against the drugs.

Acosta-Alvear et al. also found that lower levels of regulatory subunits desensitized multiple myeloma patients to therapy based on proteasome inhibition, suggesting that results from the genetic screen carried out in cells can predict clinical resistance mechanisms and guide the development of future therapies to increase patient survival.

As expected because of its essential cellular role, pharmacologic inhibition of the proteasome is inherently toxic. Consequences of proteasome inhibition leading to toxicity include the accumulation of proteasome substrates and the failure to recycle amino acids (*Suraweera et al., 2012*). Intriguingly, multiple myeloma (MM) cells are hypersensitive to proteasome inhibition, and two inhibitors of the proteolytic activity of the 20S core, bortezomib and carfilzomib, have been approved for the treatment of MM patients (*Shah and Orlowski, 2009*; *Buac et al., 2013*; *Röllig et al., 2014*). The basis for the hypersensitivity of MM cells to proteasome inhibitors is unclear. One hypothesis poses that the high protein biosynthetic rate coupled the high secretory activity of the plasma cell-like MM cells results in an increased need for clearance of misfolded proteins by the proteasome (*Meister et al., 2007*; *Bianchi et al., 2009*; *Cenci et al., 2012*). This would render these cells heavily dependent on the proteasome and the proteostasis network at large, and would account for the therapeutic window of proteasome inhibitors in the clinic.

Most MM patients initially respond to treatment with proteasome inhibitors, but the tumors eventually develop resistance (*Buac et al., 2013*). To uncover the genetic mechanisms underlying resistance to proteasome inhibitors, and to identify strategies to overcome resistance, we used our next-generation shRNA library (*Kampmann et al., 2015*) to screen for genes controlling the sensitivity and adaptation of MM cells to the proteasome inhibitor carfilzomib.

Paradoxically, we found that knockdown of 19S regulator components desensitized cells to proteasome inhibition. Previous RNAi screens had not reported this effect (*Chen et al., 2010*; *Zhu et al., 2011*), however a haploid mutagenesis screen carried out independently and in parallel to this study also found the protective effect of 19S depletion (S. Lindquist, personal communication). Lower 19S levels were also predictive of diminished response of MM patients to proteasome inhibitor-based therapy.

## Results

### Identification of genes controlling sensitivity to proteasome inhibitors

To identify genetic nodes that would delineate specific dependencies of MM cells, as well as those controlling the response of MM cells to proteasome inhibitors, we conducted an RNAi screen using our next-generation shRNA library (*Kampmann et al., 2015*). This library targets each mRNA by ~25

independent shRNAs and contains thousands of negative control shRNAs to enable robust detection of hit genes. We introduced sublibraries targeting 7712 genes involved in proteostasis, cancer, apoptosis, kinases, phosphatases and drug targets into U-266 MM cells. We then split this population into two subpopulations, each of which was grown either in the absence of drug or exposed to 1-hr pulses of carfilzomib followed by recovery (*Figure 1A*). This strategy allows for the identification of inherent vulnerabilities (*i.e.* genes affecting cell growth), as well as genes controlling sensitivity to proteasome inhibition. We identified several hundred genes that modified the response (either

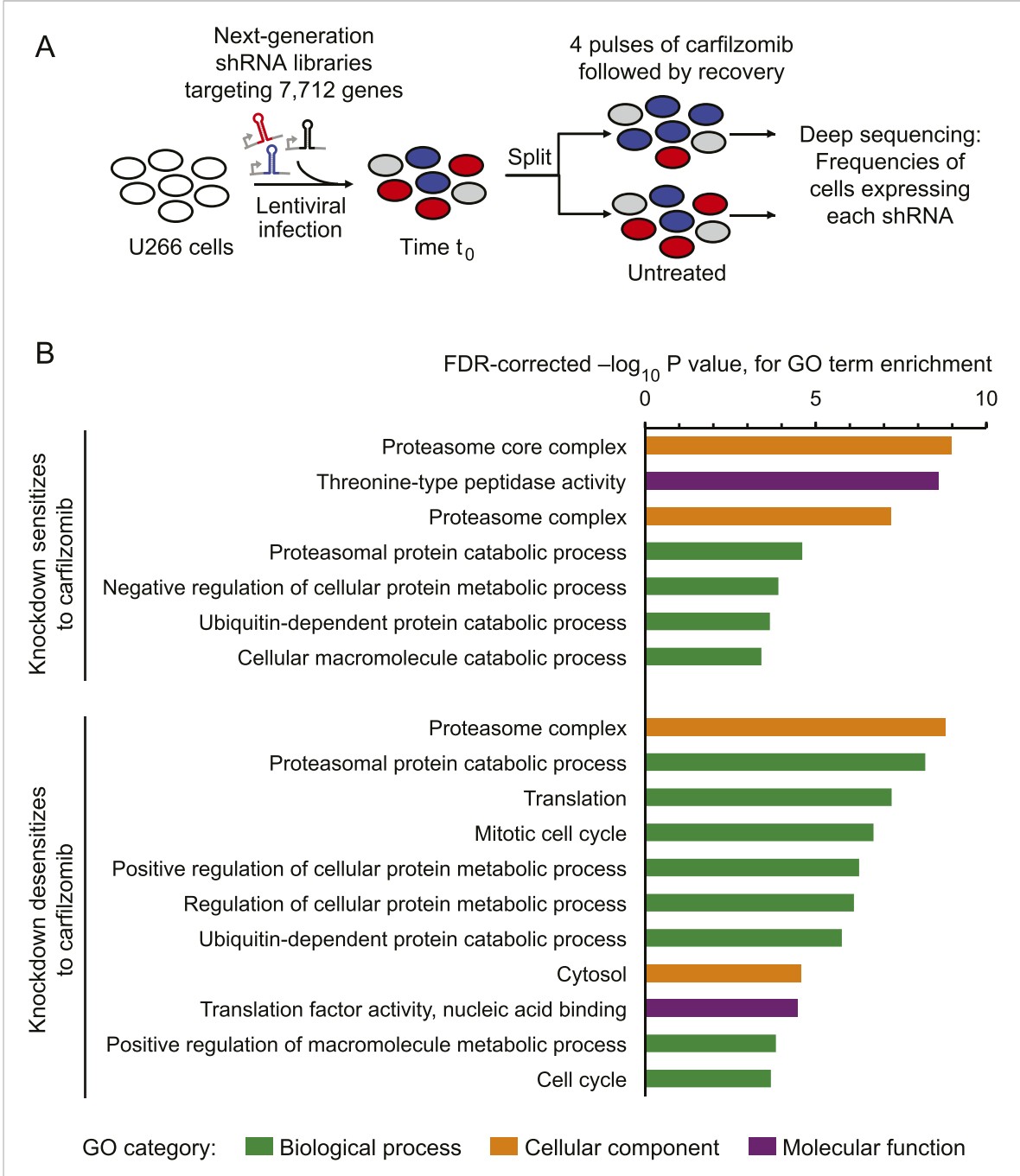

**Figure 1**. Screen for genes controlling the sensitivity of multiple myeloma cells to carfilzomib. (**A**) Screening strategy. (**B**) Gene Ontology (GO) categories enriched among the top 50 genes whose depletion results in sensitization carfilzomib and the top 50 genes whose depletion results in desensitization carfilzomib.

sensitizing or desensitizing) towards carfilzomib, as well as several hundred genes whose loss impacted cell growth (*Supplementary files 1 and 2*). Gene Ontology (GO) term enrichment analysis of the hit genes from this primary screen identified the UPS, cell cycle, and translation as major functional categories controlling the cells' response towards proteasome inhibition (*Figure 1B*).

## Nodes within the proteostasis network control the response to proteasome inhibition

As expected, the genetic depletion of the multi-drug resistance ABC transporters (ABCB1, black circle in *Figure 2A*) sensitized cells to carfilzomib. In addition, several nodes of the cytosolic proteostasis network modulated sensitivity to proteasome inhibition, including molecular chaperones (HSPA4,

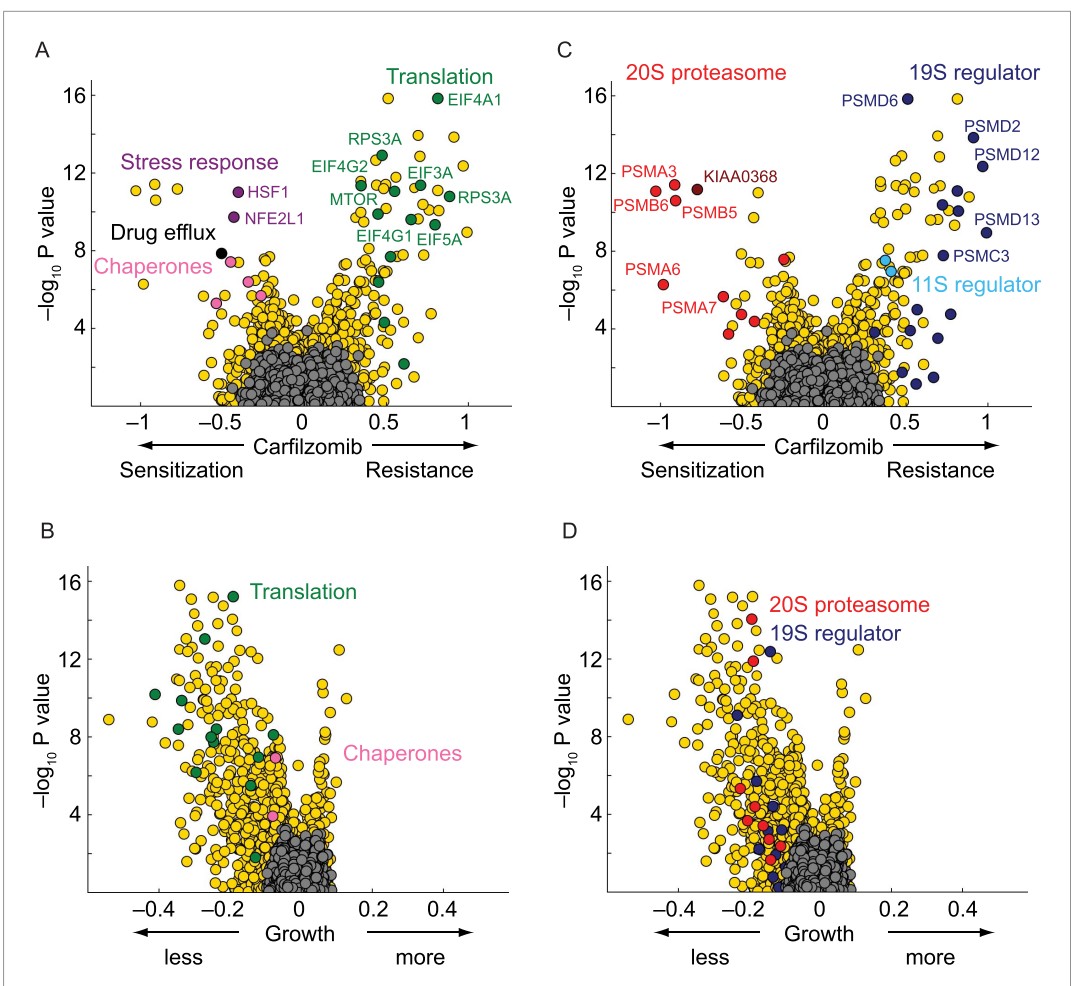

**Figure 2**. Nodes within the proteostasis network control the response of myeloma cells to carfilzomib. (**A**) Volcano plot showing knockdown effects (sensitization or desensitization to carfilzomib) and statistical significance of human genes (orange dots) and quasi-genes generated from negative control shRNAs (grey dots). Drug resistance / sensitization phenotypes were previously defined as ρ (*Kampmann et al., 2013*); a value of −1 corresponds to a twofold sensitization to the drug. Hit genes belonging to functional categories of interest are color-coded as labeled in the panels. (**B**) Volcano plot as in **A**, except showing effect on growth. Growth phenotypes were previously defined as γ (*Kampmann et al., 2013*); a value of −1 corresponds to a twofold reduction in growth rate. (**C**) Volcano plot as in **A**, highlighting the opposing effects of 20S or 19S proteasome knockdown on the sensitivity of cells towards carfilzomib. Note the protective effect is not restricted to the 19S regulator alone, but is shared with the 11S regulator. (**D**) Volcano plot as in **C**, except showing effect on growth.

The following figure supplement is available for figure 2:

**Figure supplement 1**. Comparison of growth phenotypes and carfilzomib resistance phenotypes for each targeted gene.

HSPA8, HSPA90AB1; pink circles in *Figure 2A*), and stress response transcription factors (HSF1, NFE2L1; purple circles in *Figure 2A*). Conversely, knockdown of several genes directly participating in protein synthesis conferred protection (green circles in *Figure 2A*), most notable including components of the EIF4F translation initiation complex (EIF5A, EIF4A1, EIF4E, EIF4G1, EIF4G2, EIF3A, EIF3F), as well as the elongation factor EEF2, ribosomal RNA polymerase (POLR1D), ribosomal proteins (RPS3A, RPS6, RPS25), and MTOR, the master regulator of protein synthesis, even though knockdown of these factors in the absence of carfilzomib was detrimental to cell growth (*Figure 2B*). This finding is consistent with the notion that decreased protein synthesis alleviates the load on proteasome (*Chen et al., 2010*; *Cenci et al., 2012*).

Some of the relevant nodes of the proteostasis network that we identified can be targeted pharmacologically. Based on the protective effect of MTOR knockdown, we hypothesized that its inhibition by rapamycin would desensitize cells to carfilzomib. Indeed, we observed the expected protective effect of rapamycin (*Figure 3*). Since MTOR inhibition can also induce autophagy (Reviewed in *Sarkar, 2013*), we tested whether the MTOR-independent induction of autophagy by trehalose (*Sarkar et al., 2007*) would confer similar protection. Our results support the protective role of autophagy during proteasome inhibition (*Figure 3—figure supplement 1*), indicating that MTOR inhibition may desensitize to carfilzomib both through inhibition of translation and induction of autophagy. These experiments illustrate the potential of our functional genomics approach to predict drug–drug interactions on the cellular level.

## Paradoxical phenotype of 19S knockdown

Knockdown of several subunits of the 20S proteasome core itself (PSMB1, PSMB4, PSMB5, PSMB6, PSMA2, PSMA3, PSMA7, red circles in *Figure 2C*), as well as genetic depletion of KIAA0368/ECM29 (brown circle in *Figure 2C*), an adaptor/scaffold protein that associates with the 20S core (*Gorbea et al., 2004*), and NFE2L1 (purple circle in *Figure 2A*), a transcription factor controlling proteasome biogenesis (*Radhakrishnan et al., 2010*; *Steffen et al., 2010*), provided strong sensitization to proteasome inhibition. This finding is consistent with previous studies in which depletion of a protein or pathway targeted by a drug sensitizes cells to the drug used around its EC50 (*Giaever et al., 1999*; *Matheny et al., 2013*).

Unexpectedly, the genetic depletion of the vast majority of subunits of the 19S proteasomal regulator conferred marked resistance to proteasome inhibition (blue circles in *Figure 2C*). Notably, 19S subunits were among the strongest protective hits in our screen (*Supplementary file 2*). This paradoxical effect also occurred with the depletion of PSME1 and PSME2, components of the 11S regulator (light blue circles in *Figure 2C*). While knockdown of 20S and 19S subunits in the presence of carfilzomib had opposing phenotypes, knockdown of either 19S or 20S subunits in the absence of carfilzomib negatively impacted cell growth in all cases (*Figure 2D*). However, the protective effect of 19S knockdown was not simply mediated by slowing cell growth, since knockdown of many other genes had a similar or more dramatic impact on cell growth without increasing resistance to carfilzomib (*Figure 2C,D*, *Figure 2—figure supplement 1*, *Supplementary files 1 and 2*).

The opposing protective and sensitizing effects of 19S and 20S subunit knockdown were observed for the vast majority of shRNAs targeting these genes (*Figure 4A,B*). To validate the generality of our findings, we determined bortezomib sensitivity in a batch retest of shRNAs introduced into JJN-3, U-266, and RPMI-8226 MM cells, and K-562 leukemia cells. As we observed in U-266 cells treated with carfilzomib, knockdown of 20S subunits sensitized to proteasome inhibitors, whereas knockdown of 19S subunits desensitized to proteasome inhibition across this cell line panel (*Figure 4C*). In addition, we introduced single shRNAs into MM cells targeting single subunits of either the 20S core or 19S regulator. In these experiments, expression of the individual shRNAs resulted in about twofold shifts of the dose–response curves to proteasome inhibition (*Figure 4D*), lending further strong support to the results from the genetic screen.

## Depletion of 19S regulator subunits does not desensitize the 20S core to inhibitors

To gain insights into the mechanistic basis of the protective effect of 19S knockdown, we first tested whether depletion of 19S subunits changed 20S levels or protects the 20S core from proteasome inhibitors. To this end, we compared the chymotrypsin-like protease activity of the β5 subunits of the

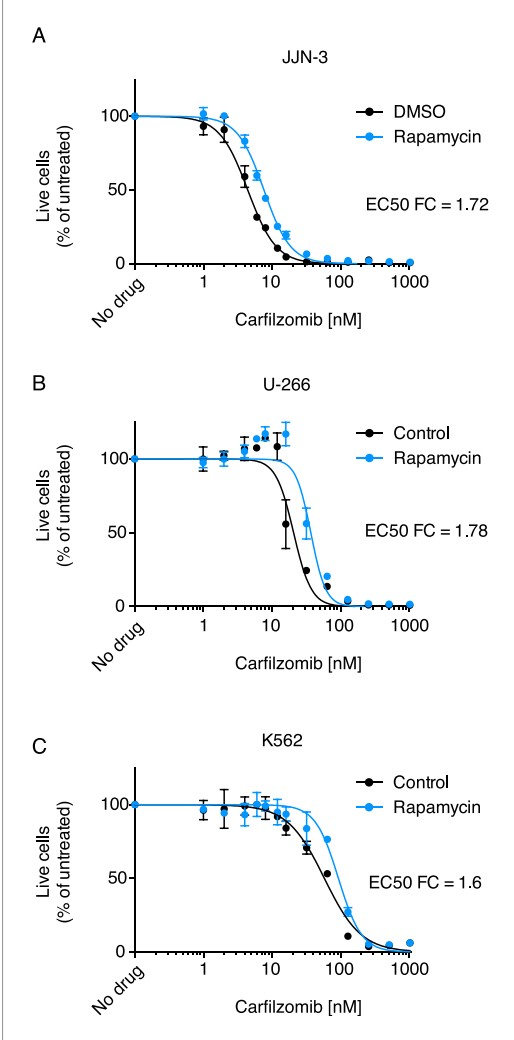

**Figure 3**. Rapamycin desensitizes cells to carfilzomib. Dose-response curves of multiple myeloma (MM) cells (**A**, **B**) and a leukemia cell line (**C**) exposed to carfilzomib after a 24 hr pretreatment with 200 nM rapamycin. FC: fold change of EC50. Data points are means of two experimental replicates, error bars denote SD.

The following figure supplement is available for figure 3:

**Figure supplement 1**. Induction of autophagy desensitizes cells to carfilzomib.

20S core (which is the direct target of carfilzomib and bortezomib) in lysates of cells expressing a negative control shRNA or an shRNA targeting the 19S subunit PSMD12 (characterized in *Figure 5—figure supplement 1*). We found that PSMD12 knockdown did not substantially increase 20S chymotrypsin-like activity, and did not affect its susceptibility to carfilzomib (*Figure 5A*). We confirmed these findings by an orthogonal approach measuring the amounts of accessible 20S subunits targeted by carfilzomib (β5 and LMP7) in a proteasome constitutive/immunoproteasome subunit enzyme-linked immunosorbent assay, ProCISE (*Parlati et al., 2009*) (*Figure 5B,C*). Together, these results suggest that the loss of 19S subunits does not lead to a decreased effectiveness of proteasome inhibitors in targeting and inhibiting the 20S core.

## Depletion of the 19S regulator subunits causes changes in the spectrum of proteasome substrates

Our results support a scenario wherein the catalytic activity of the 20S core and its sensitivity to proteasome inhibitors remains unaltered in the face of 19S depletion. Therefore, we reasoned that knockdown of 19S subunits causes profound changes to cellular physiology that desensitize cells to the effects of proteasome inhibition. Because the 19S regulator delivers substrates to the 20S catalytic core (*Liu and Jacobson, 2013*), we hypothesized that a loss in 19S function may lead to the selective accumulation of certain proteins, some of which may mitigate the effects of 20S core inhibition.

To determine the global effects of 19S knockdown on the proteome, we conducted an exploratory proteomics experiment that suggested a possible accumulation of select substrates (*Figure 6—figure supplement 1*, *Supplementary files 3 and 4*). Among the proteins accumulating upon 19S knockdown were protein degradation factors, whose accumulation we verified by quantitative immunoblot (*Figure 6*). These included SQSTM1/p62, a cargo receptor protein that delivers polyubiquitylated proteins to aggresome-like bodies degraded by the autophagosome (*Pankiv et al., 2007*), and two subunits of a heterotrimeric complex functioning in endoplasmic reticulum (ER)-associated protein degradation: UFD1L and the triple AAA ATPase VCP/p97 (*Wolf and Stolz, 2012*). Notably, these factors accumulated upon PSMD12 knockdown both in the presence and absence of carfilzomib, but not (or to a lesser degree) upon carfilzomib treatment (*Figure 6*).

By contrast, the abundance of the anti-apoptotic Bcl-2 family member MCL1, a protein rapidly turned over by the proteasome (*Schwickart et al., 2010*), was sharply increased by carfilzomib treatment, but much less affected by 19S knockdown (*Figure 6*). Similarly, 19S knockdown did not

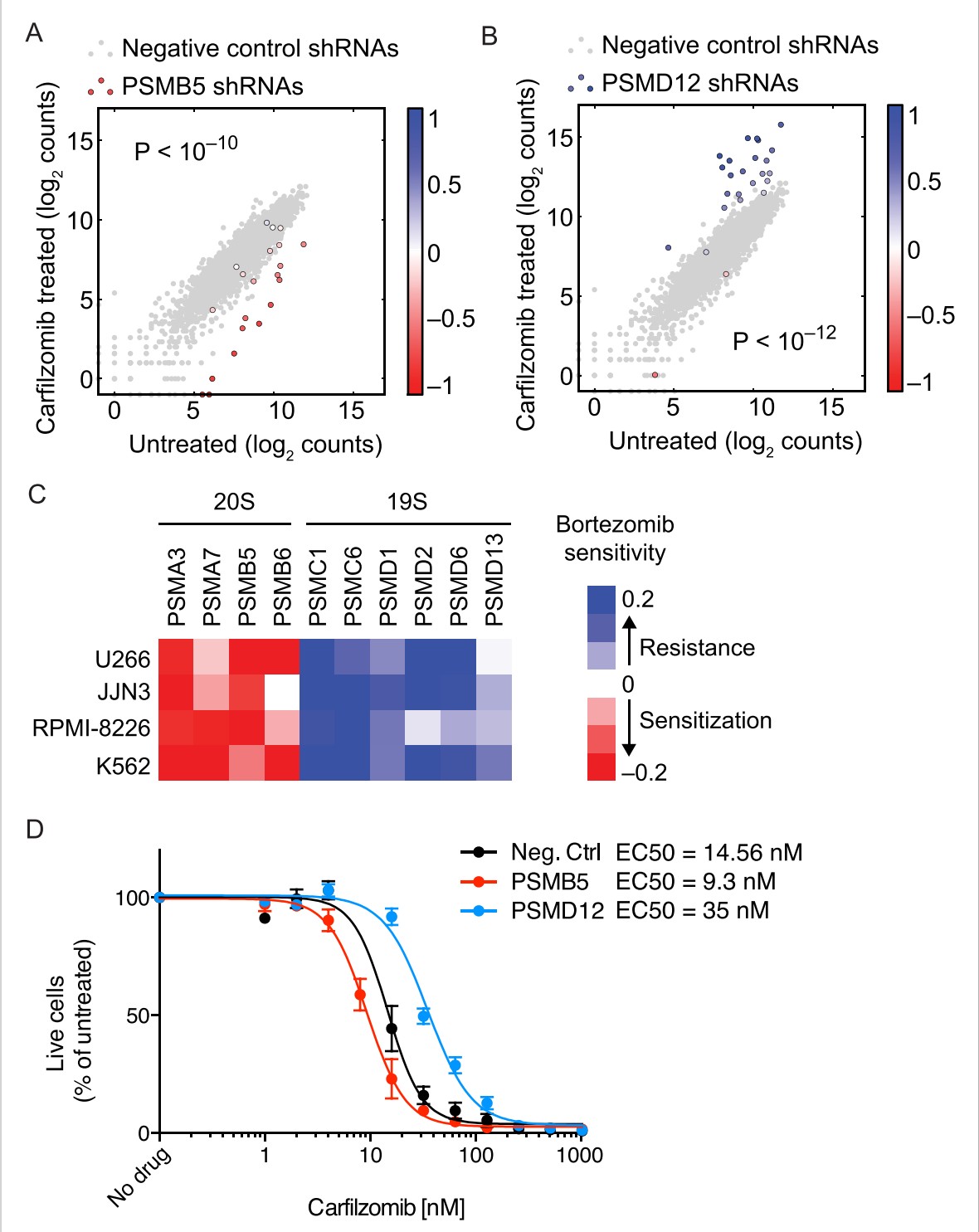

**Figure 4**. Opposing effects of 19S and 20S proteasomal subunit knockdown on carfilzomib sensitivity. (**A**, **B**) Scatter plots of the frequencies cells expressing different shRNAs targeting a 20S core subunit (**A**) or a 19S regulator subunit (**B**) in untreated or carfilzomib-treated cells. The grey dots represent cells expressing negative control shRNAs. Colored bars indicate the quantitative resistance phenotype (ρ) of each shRNA. (**C**) Heatmap showing the protective or sensitizing effect of knocking down subunits of the 19S or 20S proteasomes, respectively, in multiple cell lines. (**D**) Dose-response of U266 cells that constitutively expresses an shRNA targeting the PSMD12 subunit of the 19S proteasome, the PSMB5 subunit of the 20S proteasome, or a negative control shRNA. Data points are means of two experimental replicates, error bars denote SD.

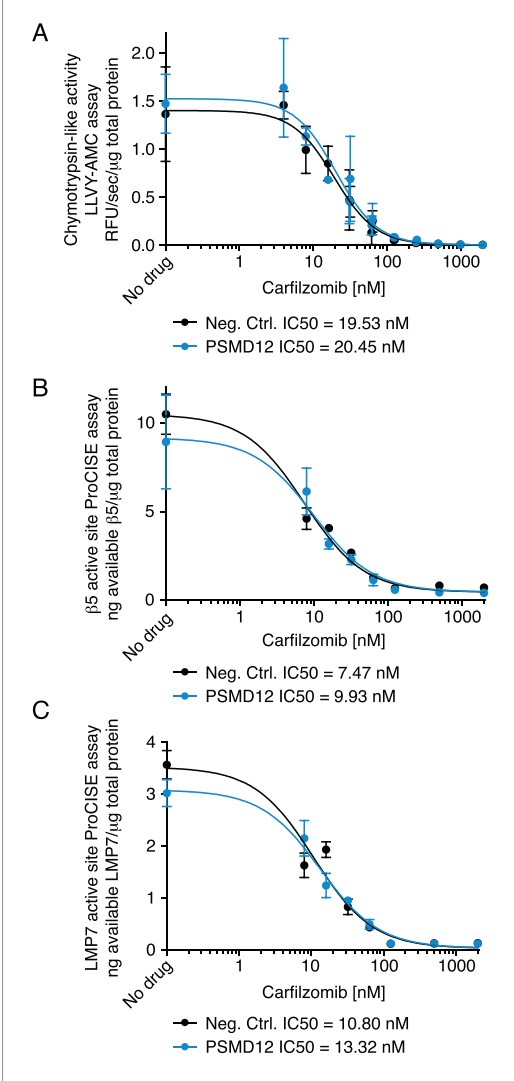

**Figure 5**. Proteasome activity in U266 cells expressing a negative control shRNA or an shRNA targeting the PSMD12 subunit of the 19S proteasome, and its susceptibility to inhibition by carfilzomib after a 1 hr treatment. (**A**) Fluorometric measurement of the chymotrypsin-like protease activity of the 20S proteasome. (**B**, **C**) Enzyme-linked immunoabsorbent assay for accessibility of the (**B**) $\beta$5 subunit and the (**C**) LMP7 subunits of the 20S proteasome. Data points are means of two experimental replicates, error bars denote SD.

The following figure supplement is available for figure 5:

**Figure supplement 1**. Characterization of samples used to measure proteasome activity in *Figure 5*.

lead to an overall enhancement of NFkB signaling, a pathway critical for survival and proliferation of lymphoid malignancies (*Demchenko and Kuehl, 2010*) (*Figure 6*). A caveat of this comparison is that the PSMD12 knockdown was constitutive, whereas the carfilzomib treatment was acute. Notwithstanding, these observations are consistent with the hypothesis that the protective phenotype we observed upon 19S knockdown may arise from the selective upregulation of protein turnover pathways and not from the upregulation of pro-survival pathways.

## The level of 19S proteasome is predictive of MM response to therapy with proteasome inhibitors

Proteasome inhibitors bortezomib and carfilzomib are used clinically to treat MM patients. Most patients respond to proteasome inhibitors to varying degrees, but eventually develop resistance. We sought to test whether the role of 19S levels in controlling sensitivity to proteasome inhibitors, which we identified in cell lines, would also be relevant in MM patients. To this end, we isolated CD138-positive cells, which include MM cells and plasma cells, from the bone marrow cells of pre-treatment MM patients. Patients then underwent carfilzomib-based combination therapy. Using previously defined criteria (*Durie et al., 2006*), we classified patients based on their response to therapy as complete responders (including complete response [CR], and stringent complete response, [sCR]) or partial responders (including partial response [PR], and very good partial response [VGPR]).

Using flow cytometry, we quantified levels of 19S regulator subunit S7 (PSMC2), 20S core subunit beta-4 (PSMB2) and aggresomes in pre-treatment CD138-positive cells. We found that 19S proteasome levels were significantly higher in the group of patients who achieved CR after treatment compared to partial responders (p < 0.0007, Mann–Whitney test; *Figure 7A*). By contrast, levels of the 20S core proteasomes were not significantly different between complete and partial responders (*Figure 7B*). Similarly, aggresome levels were not predictive of clinical outcomes (*Figure 7C*). The combination therapy used in the clinical study included lenalidomide, and while we cannot exclude an impact of 19S levels on lenalidomide response, the rapidity and depth of the response in this study (*Korde et al., 2015*) suggest that anti-tumor activity is mostly due to carfilzomib, since lenalidomide has consistently been reported to act more slowly and have a much lower CR rate than carfilzomib (*Mateos et al., 2013*).

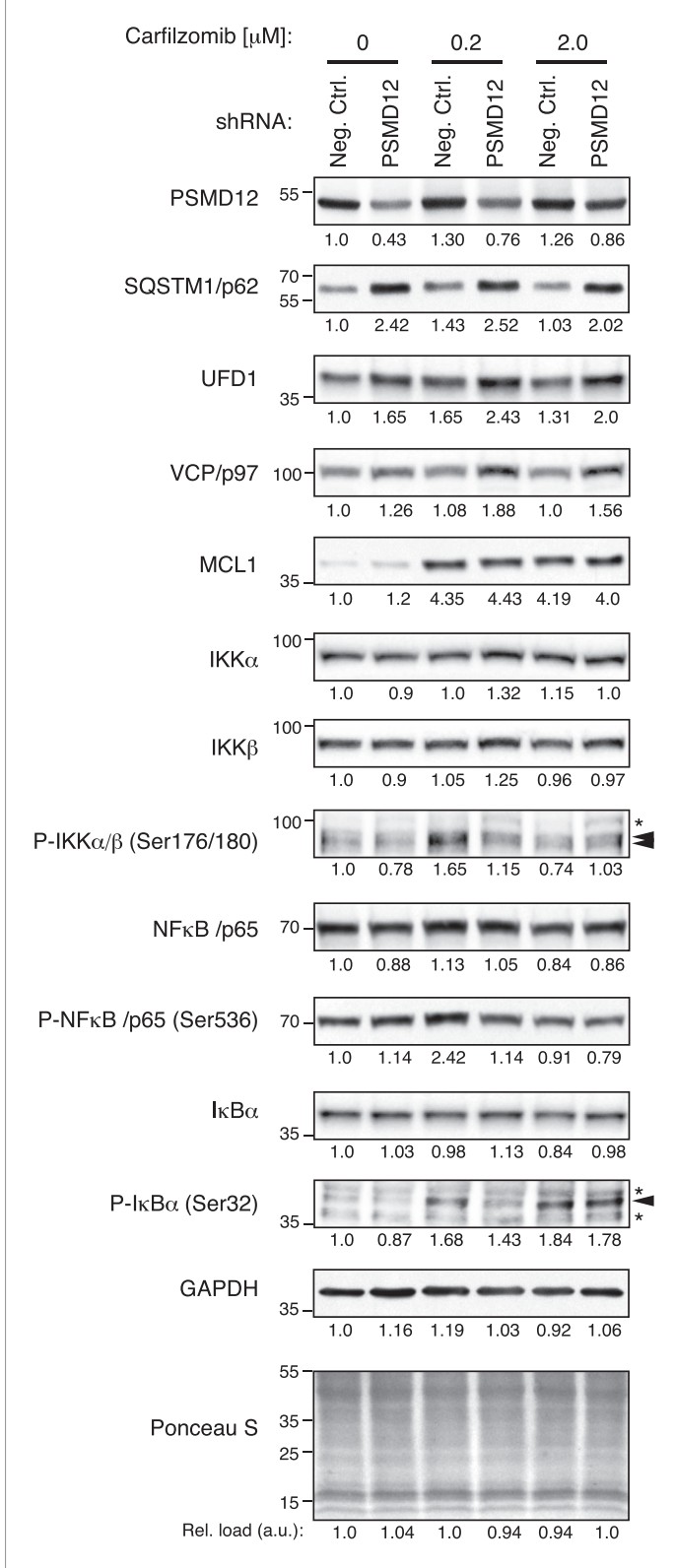

**Figure 6**. Depletion of the 19S protease regulator causes the accumulation of specific substrates. Immunoblot analysis of protein levels in U-266 cells expressing a negative control shRNA or an shRNA targeting 19S subunit PSMD12, untreated or exposed to low (200 nM) or moderate doses (2 µM) of carfilzomib for 4 hr. Numbers below the *Figure 6. continued on next page*

*Figure 6. Continued*

blots correspond to the normalized relative amount (compared to total protein in each lane). Numbers on the left margin of each panel indicate molecular weights (kDa).

The following figure supplement is available for figure 6:

**Figure supplement 1**. Determination of the global effects of 19S proteasome depletion on the proteome.

---

Taken together, these findings suggest that the desensitization to proteasome inhibition caused by decreased 19S levels is also clinically relevant in MM patients, and that 19S levels are a predictive biomarker of response to proteasome inhibitor-based therapy.

## Discussion

The exquisite sensitivity of MM cells to proteasome inhibitors provides a paradigm for non-oncogene addiction in cancer. Using our next-generation RNAi platform to identify genetic determinants of sensitivity to proteasome inhibitors, we found that incapacitating the 'executioner' (the 20S core) promotes sensitization to the pharmacological insult, while disarming the 'decision maker' (the 19S regulator) paradoxically results in resistance. In fact, knockdown of almost any 19S subunit conferred carfilzomib resistance, and together they were the group of genes with the strongest protective effect. These opposing phenotypes are consistent with our previous observations in budding yeast, where components of the 20S and 19S proteasome subcomplexes formed separate functional clusters in a genetic interaction map (*Breslow et al., 2008*).

The phenotype arising from genetic depletion of the 20S core subunits is readily understandable and in line with previous observations: diminishing the amount of the direct target of an inhibitory drug sensitizes cells to the drug (*Giaever et al., 1999*; *Matheny et al., 2013*). Two non-mutually exclusive scenarios account for this observation: silencing the 20S core subunits can lead to (i) a crippled proteasome that is easier to inhibit, or to (ii) substoichiometric amounts of the silenced subunits which in turn compromises the assembly of functional proteasomes.

The protective effect of 19S regulator knockdown was unexpected since 19S and 20S work in concert to degrade ubiquitylated protein substrates. We validated our finding in a range of MM and non-MM cells with different genetic backgrounds, as well as in MM patients, suggesting this is a general, unifying mechanism underlying the response and adaptation to proteasome inhibition. Our data supports that 19S depletion alters the spectrum of proteasome substrates, leading to selective accumulation of factors involved in protein degradation. This mechanism may represent a homeostatic feedback loop that is relevant in normal cellular contexts. In MM cells, upregulation of protein turnover pathways may reduce cellular dependence on the proteasome.

Previously reported mechanisms of resistance to proteasome inhibitors in yeast and mammalian cells include mutations or overexpression of the direct drug target, PSMB5, in the catalytic core of the proteasome (*Oerlemans et al., 2008*; *Huber et al., 2015*), but to our knowledge, they have never been identified in patients. Our results support an entirely different mechanism of drug resistance, brought about by rewiring the proteostasis network, which reduces the dependence on the proteasome. A detailed understanding of the role of the proteostasis network in disease remains a major challenge. This is due to the size of the network its dynamic nature, genetic redundancies, and context dependence (*Balch et al., 2008*). Our functional genomics approach proved to be especially well-suited to reveal functionally relevant nodes contributing to the rewiring of the proteostasis network in the context of disease.

The notion that alternative protein degradation pathways can desensitize cells to proteasome inhibitors provides a strong motivation for the simultaneous targeting of parallel pathways in combination therapy. Indeed, other groups have investigated the role of autophagy during proteasome inhibition (*Kawaguchi et al., 2011*; *Santo et al., 2012*; *Komatsu et al., 2013*; *Moriya et al., 2013*; *Mishima et al., 2015*), and combinations of bortezomib with autophagy inhibitors, including hydroxychloroquine (*Vogl et al., 2014*) and the HDAC6 inhibitor ACY-1215 (ClinicalTrail.gov identifier NCT01323751), are currently in clinical trials for MM.

While increasing protein degradation desensitizes cells to proteasome inhibition, our results also suggest and are consistent with an alternative network-level mechanism of resistance: reducing

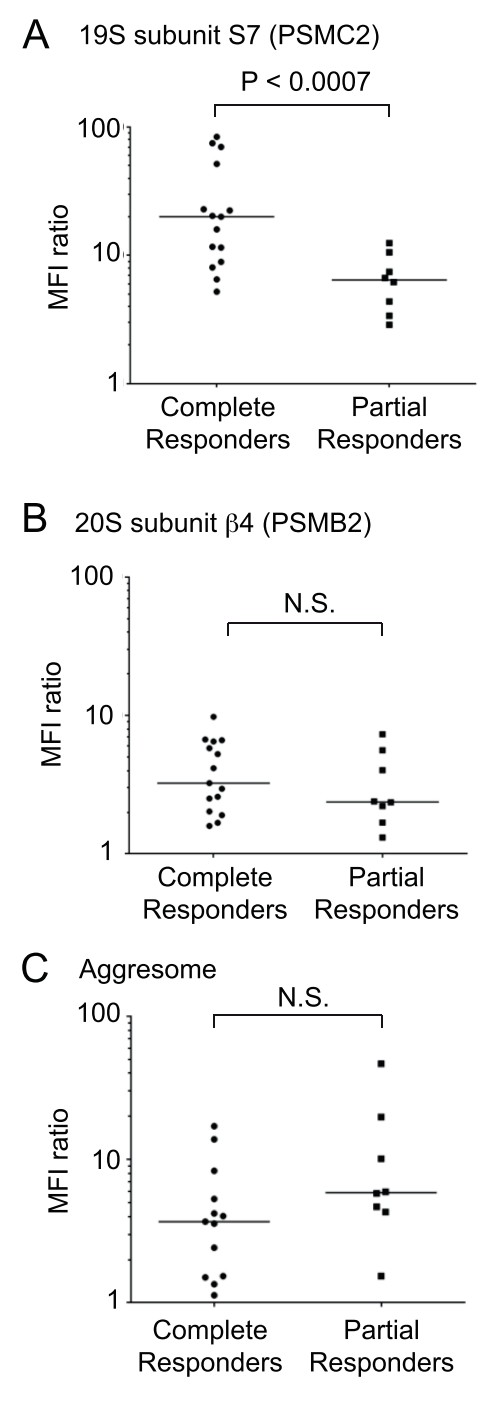

**Figure 7**. 19S proteasomal subunit levels predict the response to carfilzomib-based therapy in patients. Levels of (**A**) 19S subunit PSMC2, (**B**) 20S subunit PSMB2 and (**C**) aggresomes quantified by flow cytometry in CD138 + bone marrow cells (including plasma cells and MM cells) of patients prior to therapy in clinical trial for carfilzomib-based combination therapy. Values are shown for separately for complete responders and partial responders.

protein synthesis (*Chen et al., 2010*; *Cenci et al., 2012*). Both mTOR and the EIF4F complex are master regulators of protein synthesis, and their depletion leads to increased resistance towards proteasome inhibition. By diminishing protein synthesis, the cell could remodel its response to proteasome inhibitors, in this case not through the compensatory upregulation of alternative protein degradation pathways, but by offsetting the reduced proteasome capacity through lowered protein load.

Our functional genomics approach also pointed to other nodes in the proteostasis network that emerge as synergistic vulnerabilities with proteasome inhibition. Specifically, knockdown of the master regulator of cytosolic proteostasis, HSF1, and of individual cytosolic Hsp70 and Hsp90 chaperones sensitized cells to carfilzomib (*Figure 2A*). These nodes are therefore additional candidate targets for combination therapy with proteasome inhibitors. Hsp90 inhibitors (*Ishii et al., 2012*; *Suzuki et al., 2015*) and Hsp70 inhibitors (*Braunstein et al., 2011*) have previously been reported to act synergistically with bortezomib against MM cells.

Notably, we did not find a role for ER proteostasis factors in controlling sensitivity to carfilzomib in U-266 cells. The high secretory activity of MM cells had previously been hypothesized to challenge the folding capacity of the ER, leading to an increased burden of misfolded protein that would explain the increased dependence on the proteasome (*Meister et al., 2007*; *Bianchi et al., 2009*). A possible explanation of our findings is that during proteasome inhibition in MM cells, misfolded proteins are still extracted from the ER, as has been reported for 3-Hydroxy-3-methylglutaryl-coenzyme A reductase in the cholesterol biosynthesis pathway (*Morris et al., 2014*), and that their removal from the ER alleviates the proteotoxic load.

Our study illustrates the power of our functional genomics approach to uncover biomarkers predictive of drug responses in patients and to guide the rational design of combination therapies that show promise to overcome the urgent clinical problem of drug resistance in cancer. From a clinical perspective, an assay that can predict response to a given therapy would significantly help improving outcomes and reduce toxicities for individual patients with MM. Currently, however, therapeutic decisions are largely based on a clinical trial-and-error basis. Our results support the development of clinical assays predictive of proteasome inhibitor sensitivity, with the potential to become an essential test in every-day treatment decisions for physicians treating patients with MM. The combination of

observational genomics in patients and functional genomics in model systems can pave the way for precision medicine, while providing fundamental insights into biology.

## Materials and methods

### Cell culture and drug treatments

RPMI-8226, U-266 and JJN-3 cells were obtained from the German Collection of Microorganisms and Cell Culture (DSMZ, Braunschweig, Germany). K562 cells were a kind gift of Neil Shah (UCSF). RPMI-8226, U-266 and K562 cells were grown in RPMI medium supplemented with 10% fetal bovine serum (FBS), 4 mM L-glutamine (L-glut) and antibiotics (pen/strep). JJN-3 cells were grown in a 50:50 mixture of Iscove's Modified Dulbecco's media (IMDM) and high-glucose Dulbecco's Modified Eagle's Medium (DMEM) supplemented with 20% FBS and 4 mM L-glut and pen/strep. For proteasome inhibition studies, cells were treated (1) overnight (24 hr) with increasing doses of either carfilzomib (Onyx, South San Francisco, CA) or bortezomib (Selleck Chemicals, Houston, TX) to establish dose–response curves, or (2) for 1 hr with a concentration of the proteasome inhibitor equal to its EC50, measured 24 hr after the pulse exposure. Following the pulse exposure cells were washed twice with media and replated at a density of 0.5 million per milliliter. Exposure times and drug concentrations are indicated in the figures. To induce autophagy, cells were treated with 200 nM rapamycin (EMD Millipore, Billerica, MA) or 100 mM D-(+)-trehalose dihydrate (Sigm-Aldrich, St. Louis, MO) overnight (24 hr). After exposure to the drug, the cells were pelleted and resuspended in media without rapamycin or trehalose and supplemented with increasing concentrations of carfilzomib and incubated for an additional 24 hr to establish dose–response curves.

### Lentiviral and retroviral transductions

The retroviral expression construct pBABE-puro mCherry-EGFP-LC3B was a gift from Jayanta Debnath (Addgene, Cambridge, MA], plasmid # 22,418) (N'Diaye et al., 2009). Retroviral transductions were carried out by transfection of 16 µg of retroviral vector and 2 µg of a plasmid encoding the envelope protein VSV-G (Clontech Laboratories, Inc., Mountain View, CA) into the packaging cell line GP2-293 (Clontech) using lipofectamine 2000 (Life Technologies, Grand Island, NY). 24 hr after transfection, the cells were switched to virus collection medium (DMEM supplemented with 4% FBS, 15 mM HEPES and 2 mM L-glut). 24 hr later (48 hr after transfection), high-titer retroviral supernatant was collected, filtered through a 0.45 µm PVDF membrane and supplemented with 8 µg/ml polybrene. The supernatant was used immediately for infection of 2.5 million target cells at a density of 1 million cells per milliliter. Transduction was accomplished by spinoculating the cells at 2000 rpm. After Spinoculation, the cells were recovered, spun down and resuspended at ~500,000 cells per milliliter in the appropriate media. Lentiviral transductions were carried out as previously described (Kampmann et al., 2014).

### Primary shRNA screen

Sublibraries of our next-generation shRNA library (Kampmann et al., 2015) targeting 7712 genes involved in proteostasis, cancer, apoptosis, kinases, phosphatases and drug targets were introduced into U-266 MM cells at an MOI of ~0.3 (ensuring that most cells express at most one shRNA). Cells transduced with the shRNA libraries were selected using puromycin. This population was split into two subpopulations. One was grown in the absence of drug, whereas the other was exposed to 1 hr pulses of carfilzomib at a concentration around the LD50 measured at 24 hr (150–200 nM), followed by recovery. After 4 rounds of treatment and recovery, treated and untreated cells were harvested, genomic DNA was isolated and the shRNA-encoding cassette was PCR-amplified and subjected to next-generation sequencing as previously described (Kampmann et al., 2014). Growth and carfilzomib resistance phenotypes were quantified and p values for hit genes were calculated using gimap software (Kampmann et al., 2013, 2014) (http://gimap.ucsf.edu). Gene-based phenotypes were calculated by averaging the phenotypes of the 3 most extreme shRNAs targeting this gene, excluding shRNAs for which there were less than 50 sequencing reads in both the treated and the untreated population. This ad hoc metric is defined arbitrarily as a compromise between averaging fewer shRNAs (which may be too sensitive to outliers) and averaging too many shRNAs (which would include inactive shRNAs and thereby underestimate the effect size).

## Validation screen

An average of 3 shRNAs targeting selected hit genes were selected based on their activity in the primary screen and individually cloned to build a focused custom shRNA library. This library was introduced into U-266, JJN-3, RPMI-8226 and K562 cells. Cells were grown untreated, or treated with pulses of bortezomib, following a similar selection strategy as for the primary screen. Growth and bortezomib resistance phenotypes were quantified and phenotypes were averaged for shRNAs targeting the same gene using gimap software (Kampmann et al., 2013, 2014) (http://gimap.ucsf. edu). For individual validation experiments, a negative control shRNA (5′-CGTTCTTAGGGTGAG TAAGAAATAGTGAAGCCACAGATGTATTTCTTACTCACCCTAAGAACT-3′), an shRNA targeting PSDM12 (5′- CAGCCTTTCTCTCAAATCTAGTTAGTGAAGCCACAGATGTAACTAGATTTGAGAGAAAGGCTT-3′) or an shRNA targeting PSMB5 (5′-CGACGGTGAAGAAGGTGATAGATAGTGAAGCCACAGATGTATCTAT CACCTTCTTCACCGTCT-3′) were expressed in the lentiviral vector pMK1200 (Kampmann et al., 2013) or pMK1224 (details provided on request).

## Other bioinformatic analysis

For GO term enrichment analysis of hit genes from the primary screen, top hits were defined as follows: The 50 genes with the most sensitizing and most desensitizing gene phenotypes and a minimal p value of $10^{-4}$ were defined as top sensitizing and desensitizing hits. GO term enrichment p values were calculated using Database for Annotation, Visualization and Integrated Discovery (DAVID) (Huang et al., 2009a, 2009b), using the set of 7712 genes targeted by the sublibraries used in the primary screen as the background. Where different GO-terms encompassing the same subset of hit genes were found, only one is displayed. Similarly, where a GO term described a subset of hit genes of those described by another GO term that had a more significant p value, only the more significant GO term describing a larger set of hit genes is displayed. For GO term enrichment analysis of proteins enriched based on proteome Stable Isotope Labeling by Amino Acids in Cell Culture (SILAC) experiments, the top 50 enriched proteins for each pair-wise comparison of samples was analyzed.

## Cell viability assays

Cell viability assays were carried out using the CellTiter GLO kit (Promega Corp., Madison, WI) following manufacturer's recommendations. Raw luminescence signals were collected using a SpectraMax M5 plate reader (Molecular Devices, Sunnyvale, CA) and an integration time of 500 milliseconds. The raw counts were normalized as the percent of signal relative to untreated cells, or the percent maximum signal when comparing treatments with more than one drug. Sigmoidal dose–response curve fitting for EC50 calculation was performed using the Prism V5 package (GraphPad Software Inc., San Diego, CA).

## Immunoblotting

Total cell lysates were collected in SDS sample buffer (62.5 mM Tris pH 6.8, 10% glycerol, 2% SDS, 0.004% bromophenol blue). Lysates were sonicated for ∼15 s to shear the genomic DNA. 2-mercaptoethanol (2 ME) was added to a final concentration of 5% to the lysates just prior to boiling and loading on SDS-PAGE gels. Gel loading normalization was accomplished by one of two methods: (1) same number of live cells per condition, or (2) densitometry after SDS-PAGE electrophoresis, transfer onto nitrocellulose membranes and staining with Ponceau stain. Normalized total cell lysate amounts were loaded onto precast Tris-glycine SDS-PAGE gels (BioRad), electrophoresed, and transferred onto 2.0 μm pore nitrocellulose membranes. Equal loading per lane was verified by staining the membrane with Ponceau stain. Destained membranes were blocked for 1–2 hr in Tris-borate saline supplemented with 0.1% Tween 20 (TBST) and 5% non-fat milk. Blocked membranes were incubated overnight with primary antibodies diluted in TBST supplemented with 5% bovine serum albumin. Antibodies and dilutions were as follows: anti-SQSTM1/p62 (D5E2) rabbit monoclonal antibody (Cell Signaling Technology [Danvers, MA] #8025, 1:1000), anti-GAPDH rabbit polyclonal antibody (Abcam [Cambridge, MA] ab9485, 1:2000), anti-PSMD12 rabbit polyclonal antibody (Bethyl Laboratories Inc. [Montgomery, TX] #A303-830A, 1:2000), anti-PSMB5 rabbit polyclonal antibody (Bethyl Laboratories, Inc., Montgomery, TX, #A303-847A, 1:5000), anti-VCP/p97 rabbit polyclonal antibody (Cell Signaling Technology #2648, 1:1000), anti-MCL1 (D35A5) rabbit monoclonal antibody (Cell Signaling Technology #5453, 1:1000), anti-UFD1L rabbit polyclonal antibody (Bethyl Laboratories Inc. #A301-875A 1:500), NFkB pathway antibodies (NFkB sample kit, Cell Signaling Technology #9936, all at 1:1000). Membranes

were incubated with HRP-conjugated secondary antibodies diluted in TBST supplemented with 5% non-fat milk at a 1:5000 dilution (Amersham, GE Healthcare Life Sciences [Pittsburgh, PA] NA931, NA934) for 1 hr at room temperature. Blots were developed using luminol-based enhanced chemiluminescence substrates (SuperSignal West Dura Extended Duration Substrate, or SuperSignal West Femto Maximum Sensitivity Substrate, Life Technologies) and exposed to radiographic film or imaged directly in a digital gel imager (Chemidoc XRS+, BioRad). Digital images were automatically adjusted for contrast using the photo editor Adobe Photoshop (Adobe Systems, San Jose, CA). Densitometric quatification of immunoblots was performed using the software package ImageJ (National Institutes of Health, Bethesda, MD) (*Schneider et al., 2012*). The background subtracted area under the curve for each band was quantified and normalized to either total protein or loading controls.

## Proteasome activity assays

U-266 cells were transduced with a negative control shRNA or an shRNA targeting PSMD12 (sequences as described above). Cell populations were then split and treated with different concentrations of carfilzomib for 1 hr or left untreated. Proteasome activity was assayed in lysates of these cell populations using the ProCISE assay (*Parlati et al., 2009*) or the LLVY-AMC for chymotryptic activity (Boston Biochem, Cambridge, MA, #S-280) following the manufacturer's instructions, except that a buffer consisting of 20 mM Tris, pH 8.0, 0.5 mM EDTA without SDS was used.

## SILAC based quantification of protein levels and di-Gly-lysine modified peptides

K562 cells expressing negative control shRNA or an inducible shRNA targeting PSMD6 (as used for proteasome activity assays) were grown for 7 days in medium containing either standard lysine and arginine (light isotopes), lysine (4,4,5,5-D4) and arginine (U-13C6) (medium isotopes) or lysine (U-13C6, U-15N2) and arginine (U-13C6, U-15N4) (heavy isotopes) obtained from Cambridge Isotope laboratories (Cambridge, MA). shRNA expression was then induced for 48 hr, after which some cell populations were treated with bortezomib as described for the proteasome activity assays. Cells were combined in two triple-SILAC experiments as follows: Combination A: SILAC light: untreated negative-control; SILAC medium: bortezomib-treated ; SILAC heavy: PSMD6 knockdown and bortezomib-treated. Combination B: SILAC light: untreated negative-control; SILAC medium: PSMD6-knockdown no inhibitor; SILAC heavy: PSMD6-knockdown and bortezomib-treated. SILAC labeled cells were lysed with modified RIPA buffer (50 mM Tris–HCl pH 7.5, 150 mM NaCl, 1% Nonidet P-40, 0.1% sodium-deoxycholate, 1 mM EDTA) containing protease inhibitors (Complete protease inhibitor mixture tablets, Roche Diagnostics, Indianapolis, IN) and N-ethylmaleimide (5 mM). The lysates were incubated for 10 min on ice and subsequently cleared by centrifugation at 16,000 × g. An equal amount of protein was mixed from different SILAC states and proteins were precipitated by adding chilled acetone (final concentration 80%) and storing overnight at −20˚C. Proteins were redissolved in denaturing buffer (6 M urea, 2 M thiourea, 10 mM HEPES pH 8.0) and subsequently reduced with dithiothreitol (1 mM) and alkylated with chloroacetamide (5.5 mM). Proteins were proteolysed with Lysyl endoproteinase C (Lys-C) for 6 hr and after fourfold dilution in water with trypsin overnight. The digestion was stopped by addition of trifluoroacetic acid, incubated at 4˚C for 2 hr and resulting precipitates removed by centrifugation for 15 min at 4000 × g. Cleared peptides were purified by reversed-phase Sep-Pak C18 cartridges (Waters Corporation, Milford, MA). Di-Gly-lysine containing peptides were enriched using the Ubiquitin Remnant Motif Kit (Cell Signaling Technology), according to the manufacturer's protocol. In brief, peptides were eluted from the Sep-Pak C18 cartridges and incubated with 40 µl of anti-di-Gly-lysine antibody resin in 1X immunoaffinity purification (IAP) buffer on a rotational wheel for 4 hr at 4˚C (*Wagner et al., 2011*). After centrifugation, the supernatant (containing unbound peptides) was removed and used for proteome analysis (see below). The immunoenriched peptides were washed three times with 1X IAP buffer and two times with water. Peptides were eluted with 0.15% trifluoroacetic acid in water. Eluted peptides were fractionated into 6 fractions by micro-column-based strong-cation exchange chroma-tography (SCX) and desalted by reversed phase C18 Stage-tips. Similarly, for analysis of protein levels, unbound peptides from the anti-di-Gly-lysine enrichment (see above) were fractioned by micro-SCX and purified by reversed phase C18 Stage-tips.

## Mass spectrometry and data analysis

Peptide fractions were analyzed on a quadrupole Orbitrap (Q Exactive, Thermo Scientific, Waltham, MA) mass spectrometer equipped with a nanoflow HPLC system (Thermo Scientific). Peptides were loaded onto C18 reversed-phase columns and eluted with a linear gradient from 8% to 40% acetonitrile containing 0.5% acetic acid. The mass spectrometer was operated in a data-dependent mode, automatically switching between MS and MS/MS. Survey full scan MS spectra (m/z 300–1200) were acquired in the Orbitrap mass analyzer. The 10 most intense ions were sequentially isolated and fragmented by higher-energy C-trap dissociation (HCD). Peptides with an unassigned charge state, as well as peptides with a charge state less than +2 for proteome samples, and +3 for di-glycine-lysine enriched samples, were excluded from fragmentation. Fragment spectra were acquired in the Orbitrap mass analyzer. Raw MS data files were analyzed by the MaxQuant software version 1.4.1.1 (*Cox and Mann, 2008*). Parent ion and MS/MS spectra were searched against protein sequences from the UniProt knowledge database using the Andromeda search engine. Spectra were searched with a mass tolerance of 6 ppm in the MS mode, 20 ppm for the MS/MS mode, strict trypsin specificity allowing up to two missed cleavage sites. Cysteine carbamidomethylation was searched as a fixed modification. Amino-terminal protein acetylation, methionine oxidation and N-ethylmaleimide modification of cysteines, and di-Gly-lysine were searched as variable modifications, and di-Gly-lysines were required to be located internally in the peptide sequence. Site localization probabilities were determined by MaxQuant using a post-translational modification scoring algorithm as described previously (*Cox and Mann, 2008*). Using the target-decoy search strategy (*Elias and Gygi, 2007*) and a posterior error probability filter, a false discovery rate of less than one percent was achieved.

## Proteasome and aggresome detection in plasma cells of myeloma patients

23 patients on a clinical research protocol using carfilzomib, lenalidomide, and dexamethasone (CRd) treatment in newly diagnosed MM patients were evaluated at the National Institute of Health (NIH, Bethesda, MD). Patients were tested for the expression of 19S and 20S proteosome subunits and aggresome levels in bone marrow plasma cells before start of carfilzomib therapy. Bone marrow aspirates were collected and immunostained using antibodies against CD45, CD38, CD138 (Becton Dickinson, San Jose, CA), Proteasome 19S S7 (19S) and Proteasome Subunit Beta Type 4 (beta4) (Abcam, Cambridge, UK). The antibody used for immunodetection of the S7 subunit of the 19S proteasome was previously validated (*Rousseau et al., 2009*). In parallel, cells were labeled with ProteoStat Aggresome Detection Reagent (Enzo Life Sciences, Farmingdale, NY). Multicolor acquisition and analysis was performed using BD FACS CANTO II and DIVA software. Data was expressed as mean fluorescence intensity (MFI) ratio using isotype-matched controls. Statistical analysis was performed using Excel (Microsoft Corporation, Redmond, WA) and DataPrism (Seattle, WA) software.

## qRT-PCR

U-266 cells carrying a constitutive non-targeting shRNA or an shRNA targeting the 19S subunit PSMD12 were collected by centrifugation, washed twice with ice-cold PBS and lysed in TRIzol (Life Techologies, Grand Island, NY). RNA was extracted following manufacturer's recommendations. 500 ng of total RNA were reverse transcribed using the SuperScript VILO cDNA synthesis kit (Life Technologies) following manufacturer's recommendations. The resulting cDNA reactions were diluted 10-fold with 10 mM Tris pH 8.0 and 1% of this dilution was used for each quantitative real-time PCR (qPCR) reaction. qPCR reactions were set-up using IQ SYBR Geen Super Mix (BioRad, Hercules, CA) in 20 µl reactions. The reactions were ran on a BioRad CFX96 Real Time system (BioRad) and analyzed using the CFX Manager Software V3.0 (BioRad). All reactions were normalized to an internal loading control (GAPDH) and the fold-changes reflecting the extent of knockdown were then normalized to the no drug, negative control shRNA condition. Average values for three different oligonucleotide pairs targeting the PSMD12 trasncript were taken for the calculations. The following oligonucleotides taregting human transcripts were used: Hs_GAPDH_Fwd: 5'-AGCCACATCGCTCAGACAC-3', Hs_GAPDH_Rev: 5'-TGGAAGATGG TGATGGGATT-3', Hs_PSMD12_exon9-3'UTR_Fwd: 5'- AATGAAAAGGATGGCACAGC-3', Hs_PSMD12_exon9-3'UTR_Rev: 5'- TTTGGATCCTTGGGTCTCTG-3', Hs_PSMD12_RefSeq_Var1_Fwd: 5'- CGTCAAGATGGAGGTG GACT-3', Hs_PSMD12_RefSeq_Var1_Rev: 5'- TCCAGAGAGAGAAGGGTTTCA-3', Hs_PSMD12_exon1-3_Fwd: 5'- CGTCAAGATGGAGGTGGACT-3', Hs_PSMD12_exon1-3_Rev: 5'- AGATACGGGATGTCGA TACCA-3'

## Acknowledgements

We are grateful to Brian Tuch (then at Onyx Pharmaceuticals, an Amgen subsidiary), Marc Shuman (UCSF), Cammie Edwards (UCSF), Owen Chen (UCSF) and members of the Walter and Weissman labs for insightful discussions and technical assistance. This work was supported by the Onyx-UCSF Oncology Innovation Alliance (DAA, MK, JSW, PW), the Howard Hughes Medical Institute Collaborative Innovation Alliance (JSW, PW), NIH/NCI U01 CA168370 (JSW), an Irvington Postdoctoral Fellowship of the Cancer Research Institute (DAA), Jane Coffins Child Postdoctoral Fellowship and NIH/NCI Pathway to Independence Award K99/R00 CA181494 (MK). CC is supported by Hallas Møller Investigator and Sapere Aude research grants from the Novo Nordisk Foundation and the Danish Council for Independent research, respectively. Center for Protein Research is supported by a generous donation from the Novo Nordisk Foundation (grant number: NNF14CC0001).

## Additional information

### Competing interests

TJB: is an employee of Onyx Pharmaceuticals, an Amgen subsidiary. AGL: is an employee of Onyx Pharmaceuticals, an Amgen subsidiary. The authors declare that no competing interests exist.

### Funding

| Funder | Grant reference | Author |
|---|---|---|
| Howard Hughes Medical Institute | Collaborative Innovation Alliance | Peter Walter, Jonathan S Weissman |
| National Institutes of Health | Pathway to Independence Award K99/R00 CA181494 | Martin Kampmann |
| University of California at San Francisco | Onyx- Oncology Innovation Alliance | Martin Kampmann, Diego Acosta-Alvear, Peter Walter, Jonathan S Weissman |
| Cancer Research Institute | Irvington Postdoctoral Fellowship | Diego Acosta-Alvear |
| Jane Coffin Childs Memorial Fund for Medical Research | Postdoctoral Fellowship | Martin Kampmann |
| Novo Nordisk | Hallas Møller Investigator | Chunaram Choudhary |
| Det Frie Forskningsråd | Sapere Aude research grant | Chunaram Choudhary |
| National Institutes of Health | U01 CA168370 | Jonathan S Weissman |

The funders had no role in study design, data collection and interpretation, or the decision to submit the work for publication.

### Author contributions

DA-A, Conception and design, Acquisition of data, Analysis and interpretation of data, Drafting or revising the article; MYC, TW, TJB, AGL, Acquisition of data, Analysis and interpretation of data; OS, JH, Acquisition of data; NK, OL, IM, CC, Conception and design, Analysis and interpretation of data; PW, Conception and design, Drafting or revising the article; JSW, Conception and design, Drafting or revising the article, Contributed unpublished essential data or reagents; MK, Conception and design, Acquisition of data, Analysis and interpretation of data, Drafting or revising the article, Contributed unpublished essential data or reagents

### Author ORCIDs

Martin Kampmann, http://orcid.org/0000-0002-3819-7019

### Ethics

Clinical trial registration NCT01402284
Human subjects: The registered clinical research trial (NCT01402284) was approved by the National Cancer Institute (NCI) Institutional Review Board (IRB) and complied with the Declaration of Helsinki,

the International Conference on Harmonization, and the Guidelines for Good Clinical Practice. All enrolled patients meeting criteria were consented with an IRB-approved document.

## Additional files

### Supplementary files

• Supplementary file 1. Growth phenotypes from shRNA screen.

• Supplementary file 2. Carfilzomib phenotypes from shRNA screen.

• Supplementary file 3. Proteome changes from SILAC experiments.

• Supplementary file 4. Ubiquitylome changes from SILAC experiments.

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
