## [Decision Letter]

Thank you for submitting your work entitled “Paradoxical resistance of multiple myeloma to proteasome inhibitors by decreased levels of 19S proteasomal subunits” for peer review at *eLife*. Your submission has been favorably evaluated by Sean Morrison (Senior Editor), and three reviewers, one of whom is a member of our Board of Reviewing Editors.

The reviewers have discussed the reviews with one another, and the Reviewing Editor has drafted this decision to help you prepare a revised submission.

This is a very interesting paper that makes the claim that knockdown of 20S proteasome subunits sensitizes cells to the proteasome inhibitor carfilzomib (not surprising), whereas knockdown of 19S subunits renders them moderately resistant to CFZ (highly unexpected). Based on a set of functional investigations into mechanism, the authors conclude that knockdown of 19S subunits upregulates autophagy, and thus the load that is normally impinging on the proteasome is redirected to a different pathway. If these claims are fully substantiated, the paper is obviously of considerable importance as it would shed new light on regulatory mechanisms that link the proteasome to other pathways for protein homeostasis.

There is agreement among the reviewers that knockdown of 19S subunits does indeed appear to render cells modestly resistant to carfilzomib, whereas knockdown of 20S subunits has the opposite effects. That said, the reviewers are not persuaded by the claim that this is due to selective upregulation of autophagy upon diminution of 19S activity, but not upon diminution of 20S activity. The reviewers are of the opinion that the authors have not adequately ruled out more mundane interpretations of the core observation that emerged from the RNAi screen. In addition, the proposed upregulation of autophagy is based on indirect analysis (protein levels) and not direct measurement of autophagic flux.

The following points, requiring additional experiments, need to be addressed in a revised manuscript. Other points listed are either not central or are important but could be addressed through text revisions (minor points).

Major points:

1) In Figure 3, the authors argue that bortezomib inhibits 20S to the same extent in control and PSMD6-depleted cells. However, the effect is monitored far outside the range of linear response and thus the experiment needs to be repeated as a comparative titration of proteasome inhibitor in control vs. PSMID12 (see minor point #1) knock-down cells to confirm that β5 inhibition is indeed similar. It is especially critical to do this as a titration because Figure 2 shows that 19S knockdown confers only a 2-fold resistance. Additional specific advice for this experiment is suggested in minor point #1.

2) The evidence that knockdown of 19S induces bypass mechanisms is based on mass spec proteomic data in Figure 4 and western data in Figure 5. Unfortunately, these experiments are comparisons of a long time-course knockdown of 19S with a short time-course drug treatment with bortezomib. Both sets of experiments need to be done as 20S depletion vs. 19S depletion. There are additional specific concerns with these experiments as well:

a) Given the focus on using knockdown of PSMD6 for the proteomics experiments, the authors should include an experiment like Figure 2 to evaluate the degree of resistance conferred by siPSMD6. The switching between different genes used for the knockdown experiments in different figures makes it difficult to evaluate consistency.

b) It is difficult to know how much weight to place on Figure 4 without seeing a comparison of biological replicates, especially given that mass spec data of this sort can be quite noisy.

c) In Figure 5, the exposure to drug was radically different from the other experiments, particularly the screen. Given that the intent is to explain the results of the screen, the experiment should be done similarly to how the screen was done, and compare fraction of red vs. green cells at the end. It is clear from the literature (e.g. PMID 20460535) that continuous exposure vs. pulsatile exposure appears to yield a substantially different set of sensitizing and desensitizing factors, and so tests of the mechanistic hypothesis should mirror the conditions used to make the core observation. Alternatively this experiment can be removed if stronger data is developed (e.g. see d) below).

d) Stronger (i.e. direct) evidence is needed to establish that depletion of 19S actually causes an increase in autophagy.

Minor points:

1) Relating to major point #1, a superior design for the experiment in Figure 3 would be to treat control and siPSMD12 cells with CFZ at a range of concentration spanning the EC50 for blocking β5 activity in the control cells, washing away the CFZ, and then measuring β5 activity in lysates prepared from these cells. This would control for a variety of potential effects on CFZ uptake, metabolism, efflux, etc. CFZ is a better agent to use than BTZ because it is completely irreversible and so the β5 activity measured in dilute lysate should accurately reflect the β5 activity that was present in the cells. Another issue is, why was PSMD6 used for this particular experiment? From the volcano plot in Figure 2 it exhibits one of the weakest effects of all the 19S subunits (albeit with one of the most highly significant P values). In addition, there is no direct comparison of PSMD6 knockdown vs. control, so it is unclear how big a change in CFZ EC50 is caused by PSMD6 knockdown (this was only done for PSMD12 in Figure 2). A target like PSMD12 that exhibits a strong effect and has been characterized in a head-to-head comparison would be an appropriate choice.

2) For the primary screen, the Methods state that the phenotypes are based on the 3 most extreme shRNAs for a given gene. What is the rationale for this? Doesn't it emphasize potential effects of outliers (e.g. off target)?

3) The way the following sentence is worded is somewhat misleading: “We validated our finding in a range of MM and non-MM cells with different genetic backgrounds, as well as in MM patients, suggesting this is a general, unifying mechanism underlying the response and adaptation to proteasome inhibition.” Although 'omib resistance mutations in β5 can be readily identified in culture, they have never been identified in a patient.

4) In Figure 2, can the authors provide a brief summary for how the sensitivity value reported on the X-axis is calculated? How big of an effect is implied by a value of 0.5 or 1.0?

5) In Figures 3 and 4, was 100 µM BTZ really used? If so, the authors should discuss the possible concern about off-target inhibition of Ser-proteases.

6) In Figure 5, what is the extent of p62 overexpression or induction by rapamycin relative to that achieved by 19S knockdown (western blot comparison)?

7) In Figure 5, it is unclear what is being reported. Shouldn't PSMD6 KD (bortezomib) and Bortezomib (PSMD6 KD) yield reciprocal results? They don't appear to, especially for SQSTM1.

8) It is unclear how an increase in VCP/p97 complex would generate resistance to PIs. Isn't this pathway still dependent on proteasome core activity for protein turnover. Dose this pathway feed other protein degradation pathways? The authors should clarify.

9) Based on Figure 5, it is argued that induction of autophagy with rapamycin confers resistance to proteasome inhibition. However, rapamycin also inhibits translation, which is well-known to increase resistance to proteasome inhibitors. Given this, these data should probably be removed because they don't actually bear on the issue at hand in an unambiguous manner.

10) While the clinical observations are extremely interesting and potentially important for therapy, it should be borne in mind that patients were also treated with lenalidomide, which induces the degradation of Ikaros by the proteasome. Is the improved response in patients with high levels of 19S due to enhanced activity of carfilzomib on the proteasome or enhanced activity of lenalidomide in mediating degradation of Ikaros? In addition, there is no information on assay and antibody validation to lend confidence that it is indeed S7 that is being measured. The authors should consider either removing these data or injecting an appropriate note of caution as to the serious limitations to their interpretation.

11) Figure 6: Please color-code the individual datapoints so that it is evident which patients are CR vs. sCR and PR vs. VGPR.

12) Figure 6: If the 19S level decrease in PR vs CR patients is relevant to the sensitization mechanism proposed in this manuscript, one would have expected to find an increase in other protein degradation pathways in the PR subset, e.g., aggresome levels, which was not observed.

13) The referencing is incomplete or incorrect in several places, including those mentioned below. These should be corrected.

a) The manuscript does not discuss or cite a prior RNAi study aimed at identifying determinants of bortezomib sensitivity and resistance (PMID 20460535).

b) The Bianchi hypothesis cited by the authors was subsequently modified, with the primary determinant being total protein synthetic rate divided by proteasome capacity (PMID: 22685320) (Introduction, second paragraph).

c) The notion that decreased translation reduces toxicity of proteasome inhibitors is presented as a novel finding of this paper, but in fact this has been reported previously in Cenci et al. (PMID: 22685320) as well as in references cited in PMID 20460535 (e.g. Discussion, fifth paragraph).

d) The role of NFE2L1 in proteasome biogenesis was actually discovered by Radhakrishnan et al. (PMID:20385086). Similar observations for the human TCF11 gene were subsequently reported by Kruger et al. (PMID:20932482).

e) It has been reported that, for HMG-CoA reductase, proteasome inhibition can block downstream of ER membrane extraction. (PMID:24860107) (Discussion, eighth paragraph).

---

## [Author Response]

*Major points*:

*1)*
Figure 3*: the authors argue that bortezomib inhibits 20S to the same extent in control and PSMD6-depleted cells. However, the effect is monitored far outside the range of linear response and thus the experiment needs to be repeated as a comparative titration of proteasome inhibitor in control vs. PSMID12 (see minor point #1) knock-down cells to confirm that β5 inhibition is indeed similar. It is especially critical to do this as a titration because*
Figure 2
*shows that 19S knockdown confers only a 2-fold resistance. Additional specific advice for this experiment is suggested in minor point #1*.

We agree with the reviewers, and we repeated the experiment over a range of carfilzomib concentrations in control vs. PSMD12 knockdown cells. In the revised Figure 5, we show that chymotrypsin-like activity of the proteasome is unchanged in cells expressing negative-control versus PSMD12-targeted shRNAs in the absence of carfilzomib, and that its inhibition by carfilzomib is indistinguishable between negative control and PSMD12 knockdown cells. In the revised Figure 5, we show that the levels of active proteasome subunits targeted by carfilzomib, β5 and its immunoproteasome version LMP7, are unchanged and equally accessible to drug in negative control and PSMD12 knockdown cells. The results support our current hypothesis: inhibition of proteasome activity remains unchanged in cells in which PSMD12 is knocked down.

*2) The evidence that knockdown of 19S induces bypass mechanisms is based on mass spec proteomic data in*
Figure 4
*and western data in*
Figure 5*. Unfortunately, these experiments are comparisons of a long time-course knockdown of 19S with a short time-course drug treatment with bortezomib. Both sets of experiments need to be done as 20S depletion vs. 19S depletion. There are additional specific concerns with these experiments as well*:

*a) Given the focus on using knockdown of PSMD6 for the proteomics experiments, the authors should include an experiment like*
Figure 2
*to evaluate the degree of resistance conferred by siPSMD6. The switching between different genes used for the knockdown experiments in different figures makes it difficult to evaluate consistency.*

*b) It is difficult to know how much weight to place on*
Figure 4
*without seeing a comparison of biological replicates, especially given that mass spec data of this sort can be quite noisy*.

*c) In*
Figure 5*, the exposure to drug was radically different from the other experiments, particularly the screen. Given that the intent is to explain the results of the screen, the experiment should be done similarly to how the screen was done, and compare fraction of red vs. green cells at the end. It is clear from the literature (e.g. PMID 20460535) that continuous exposure vs. pulsatile exposure appears to yield a substantially different set of sensitizing and desensitizing factors, and so tests of the mechanistic hypothesis should mirror the conditions used to make the core observation. Alternatively this experiment can be removed if stronger data is developed (e.g. see d) below)*.

*d) Stronger (i.e. direct) evidence is needed to establish that depletion of 19S actually causes an increase in autophagy*.

We agree with the reviewers challenging our interpretation of the SILAC results and follow-up experiments. We performed some of the suggested follow-up experiments but found the results inconclusive due to the toxicity of proteasome subunit knockdown. As such, we can currently not exclude alternative explanations of our findings and have greatly de-emphasized this part of our work in the revised version of our manuscript. We also point out all relevant caveats.

*Minor points*:

*1) Relating to major point #1, a superior design for the experiment in*
Figure 3
*would be to treat control and siPSMD12 cells with CFZ at a range of concentration spanning the EC50 for blocking β5 activity in the control cells, washing away the CFZ, and then measuring β5 activity in lysates prepared from these cells. This would control for a variety of potential effects on CFZ uptake, metabolism, efflux, etc. CFZ is a better agent to use than BTZ because it is completely irreversible and so the β5 activity measured in dilute lysate should accurately reflect the β5 activity that was present in the cells. Another issue is, why was PSMD6 used for this particular experiment? From the volcano plot in*
Figure 2
*it exhibits one of the weakest effects of all the 19S subunits (albeit with one of the most highly significant P values). In addition, there is no direct comparison of PSMD6 knockdown vs. control, so it is unclear how big a change in CFZ EC50 is caused by PSMD6 knockdown (this was only done for PSMD12 in*
Figure 2*). A target like PSMD12 that exhibits a strong effect and has been characterized in a head-to-head comparison would be an appropriate choice*.

We agree and have carried out this experiment. The results are consistent with our original hypothesis and are shown in the revised Figure 5.

2) For the primary screen, the Methods state that the phenotypes are based on the 3 most extreme shRNAs for a given gene. What is the rationale for this? Doesn't it emphasize potential effects of outliers (e.g. off target)?

The volcano plots integrate two types of information: a P value to quantify the confidence in a given hit gene (based on integrating information on all 25 shRNAs targeting that gene) and a phenotype as an ad-hoc metric reflecting the direction and size of the effect. This ad-hoc metric is defined arbitrarily as the average of the 3 most extreme shRNAs, as a compromise between averaging even fewer shRNAs (which, as the reviewers point out, may be too sensitive to outliers) and averaging too many shRNAs (which would include inactive shRNAs and thereby underestimate the effect size). shRNAs with low numbers of sequencing reads in both treated and untreated populations are in fact excluded from the set of top 3 most extreme shRNAs from which the average phenotype is calculated to provide an additional safeguard against outliers. While the phenotype is arbitrarily defined, we have found it in practice to be a useful metric. See also our use of this definition in our recent CRISPRi/a paper (PMID 25307932). We have clarified the rationale outlined above in the revised manuscript.

*3) The way the following sentence is worded is somewhat misleading: “We validated our finding in a range of MM and non-MM cells with different genetic backgrounds, as well as in MM patients, suggesting this is a general, unifying mechanism underlying the response and adaptation to proteasome inhibition.” Although 'omib resistance mutations in β5 can be readily identified in culture, they have never been identified in a patient*.

We clarified this in the revised text.

*4) In*
Figure 2*, can the authors provide a brief summary for how the sensitivity value reported on the X-axis is calculated? How big of an effect is implied by a value of 0.5 or 1.0?*

We clarified this in the revised text (Figure legend 2).

*5) In*
Figures 3 and 4*, was 100 µM BTZ really used? If so, the authors should discuss the possible concern about off-target inhibition of Ser-proteases*.

As described above, we repeated the experiment measuring proteasome activity and subunit occupancy using a range of carfilzomib concentrations (revised Figure 5). With respect to the SILAC experiments, the concern is valid and we have de-emphasized these experiments in the revised manuscript, and instead focus on immunoblot measurements (revised Figure 6). These experiments support the hypothesis that 19S knockdown blocks the degradation of specific protein degradation factors, which are not accumulating with acute 20S inhibition. For the immunoblot experiments, we used low and moderate levels of proteasome inhibitor, corresponding to ∼EC50 and 10 x EC50 for the cell line used. Even if these concentrations inhibit serine proteases, the unique effects of 19S knockdown remain a valid observation.

*6) In*
Figure 5*, what is the extent of p62 overexpression or induction by rapamycin relative to that achieved by 19S knockdown (western blot comparison)?*

As outlined in response to major point 2C, we have removed these experiments since they were not conclusive (continuous vs. pulse drug exposure).

*7) In*
Figure 5*, it is unclear what is being reported. Shouldn't PSMD6 KD (bortezomib) and Bortezomib (PSMD6 KD) yield reciprocal results? They don't appear to, especially for SQSTM1*.

We apologize for the ambiguity in the original version of this figure. No, these are independent effects. We have clarified this in the revised figure legend. All of the SILAC data have been moved to the supplement and de-emphasized. Briefly, PSMD6 KD (bortezomib) refers to the effect of PSMD6 KD in the context of bortezomib treated samples (i.e. log2 ratio of PSMD6 KD+bortezomib over ctrl KD+bortezomib), whereas bortezomib (PSMD6 KD) refers to the effect of bortezomib treatment in the context of PSMD6 KD cells (i.e. the log2 ratio of PSMD6 KD+ bortezomib over PSMD6 KD+DMSO).

*8) It is unclear how an increase in VCP/p97 complex would generate resistance to PIs. Isn't this pathway still dependent on proteasome core activity for protein turnover. Dose this pathway feed other protein degradation pathways? The authors should clarify*.

As we clarify in the revised text, each of the factors building up upon 19S knockdown not necessarily protects individually against proteasome inhibition – but rather, it more likely is the overall pattern of proteome changes that has a net protective effect. We agree with the reviewers that increases in the VCP/p97 complex alone may not protect against proteasome inhibition. However, it may be less toxic for cells to extract misfolded ER proteins to the cytosol (even if they cannot be efficiently degraded there) rather than accumulating them in the ER.

*9) Based on*
Figure 5*, it is argued that induction of autophagy with rapamycin confers resistance to proteasome inhibition. However, rapamycin also inhibits translation, which is well-known to increase resistance to proteasome inhibitors. Given this, these data should probably be removed because they don't actually bear on the issue at hand in an unambiguous manner*.

We agree and have clarified in the text that rapamycin may act both through the inhibition of protein translation and the induction of autophagy. We also now present the rapamycin data in a different context in the paper, namely to illustrate the potential of our functional genomics approach to predict drug-drug interactions. To provide independent support for the role of autophagy, we conducted an orthogonal experiment in which we induced autophagy in an mTOR-independent manner using trehalose (shown in revised Figure 3—figure supplement 1).

*10) While the clinical observations are extremely interesting and potentially important for therapy, it should be borne in mind that patients were also treated with lenalidomide, which induces the degradation of Ikaros by the proteasome. Is the improved response in patients with high levels of 19S due to enhanced activity of carfilzomib on the proteasome or enhanced activity of lenalidomide in mediating degradation of Ikaros? In addition, there is no information on assay and antibody validation to lend confidence that it is indeed S7 that is being measured. The authors should consider either removing these data or injecting an appropriate note of caution as to the serious limitations to their interpretation*.

Regarding antibody validation, this is a commercially available antibody used in multiple publications so far. It has been demonstrated that this antibody detects a 47 kDa protein representing proteasome 19S subunit S7 from HeLa cell lysate. We added this reference to the Materials and methods section describing antibody (Rousseau E et al., PMID 18986984).

With respect to the questions whether patient responses were mostly due to carfilzomib or to lenalidomide, the rapidity of the observed deep responses in the clinical study (Korde et al., PMID 26181891) suggests that carfilzomib is the driving factor, since lenalidomide + dexamethasone therapies have previously been reported to be slow-acting, with a much lower rate of complete response (Mateos et al., PMID 23902483). However, in the revised manuscript, we point out the caveat pointed out by the reviewer.

*11)*
Figure 6*: Please color-code the individual datapoints so that it is evident which patients are CR vs. sCR and PR vs. VGPR*.

Given the small sample sizes for 2 of the 4 groups, this data is hard to interpret. However we included it for the reviewers (Figure 8).

Author response image 1.**DOI:**
http://dx.doi.org/10.7554/eLife.08153.020

*12)*
Figure 6*: If the 19S level decrease in PR vs CR patients is relevant to the sensitization mechanism proposed in this manuscript, one would have expected to find an increase in other protein degradation pathways in the PR subset, e.g., aggresome levels, which was not observed.*

Aggresomes represent a degradation intermediate. Depending on the exact changes in rates of different steps in the pathway, the levels of aggresomes may not necessarily change, even if net flux through the pathway changes.

*13) The referencing is incomplete or incorrect in several places, including those mentioned below. These should be corrected*.

*a) The manuscript does not discuss or cite a prior RNAi study aimed at identifying determinants of bortezomib sensitivity and resistance (PMID 20460535)*.

b) The Bianchi hypothesis cited by the authors was subsequently modified, with the primary determinant being total protein synthetic rate divided by proteasome capacity (PMID: 22685320) (Introduction, second paragraph).

*c) The notion that decreased translation reduces toxicity of proteasome inhibitors is presented as a novel finding of this paper, but in fact this has been reported previously in Cenci et al. (PMID: 22685320) as well as in references cited in PMID 20460535 (e.g. Discussion, fifth paragraph)*.

*d) The role of NFE2L1 in proteasome biogenesis was actually discovered by Radhakrishnan et al. (PMID:20385086). Similar observations for the human TCF11 gene were subsequently reported by Kruger et al. (PMID:20932482)*.

*e) It has been reported that, for HMG-CoA reductase, proteasome inhibition can block downstream of ER membrane extraction. (PMID:24860107) (Discussion, eighth paragraph)*.

Thank you for suggesting these references, we incorporated them into the revised manuscript.